# Surface Modification of Bi_2_O_3_ Nanoparticles with Biotinylated β-Cyclodextrin as a Biocompatible Therapeutic Agent for Anticancer and Antimicrobial Applications

**DOI:** 10.3390/molecules28083604

**Published:** 2023-04-20

**Authors:** Jogy Alex, Thomas V. Mathew

**Affiliations:** Department of Chemistry, St. Thomas College Palai, Arunapuram P.O., Kottayam 686574, Kerala, India

**Keywords:** bismuth oxide nanoparticles, beta-cyclodextrin, biotin, antibacterial, anticancerous studies

## Abstract

Bismuth oxide nanoparticles with appropriate surface chemistry exhibit many interesting properties that can be utilized in a variety of applications. This paper describes a new route to the surface modification of bismuth oxide nanoparticles (Bi_2_O_3_ NPs) using functionalized beta-Cyclodextrin (β-CD) as a biocompatible system. The synthesis of Bi_2_O_3_ NP was done using PVA (poly vinyl alcohol) as the reductant and the Steglich esterification procedure for the functionalization of β-CD with biotin. Ultimately, the Bi_2_O_3_ NPs are modified using this functionalized β-CD system. The particle size of the synthesized Bi_2_O_3_ NPs is found to be in the range of 12–16 nm. The modified biocompatible systems were characterized using different characterization techniques such as Fourier transform infrared spectroscopy (FTIR), transmission electron microscopy (TEM), scanning electron microscopy (SEM), X-ray powder diffraction (XRD) and Differential Scanning Calorimetric analysis (DSC). Additionally, the antibacterial and anticancerous effects of the surface-modified Bi_2_O_3_ NP system were also investigated.

## 1. Introduction

Inorganic nanoparticles (NPs) have achieved great significance in recent times for their potential to be used in biological and therapeutic applications [1,2,3,4,5,6,7,8]. The key characteristics of nanoparticles include (i) the high-level surface area to volume ratio of NPs which enables the inclusion of a high density of functional moieties [9], (ii) the distinctive physicochemical characteristics (electric, optical, and magnetic) resulting from the nanoscale scale [10], and (iii) NPs can interact with cells and biomolecules more effectively because of their similar size to biomacromolecules [11,12], and these can be explored to open the myriad possibilities ranging from therapeutic practices to space technologies [13,14]. There are some nanoparticles and their composites, although that is not as well known. This paper focuses on the Bi_2_O_3_ nanoparticles that belong to this category. These nanoparticles raised more and more interest in the scientific community due to their cost-effective manufacturing processes, high stability, and versatility in terms of their size and shape. Moreover, the high atomic number of bismuth brings about high energy radiation attenuation greater than that of lead, making it at negligible risk of toxicity. Bismuth oxide has high dielectric permittivity, good refractive index, photoconductivity, and photoluminescence [15]. Thus, it has potential uses in a variety of fields, including solid-state fuel cells [16], photovoltaic cells [17], high-temperature superconductors [18], photocatalysts [19], gas sensors [20], optical coatings [21], and biomedical applications [22]. The five crystallographic polymorphs of Bi_2_O_3_ are α-Bi_2_O_3_ (monoclinic), β-Bi_2_O_3_ (tetragonal), γ-Bi_2_O_3_ (body-centered cubic), δ-Bi_2_O_3_ (cubic), and ε-Bi_2_O_3_ (triclinic). α-Bi_2_O_3_ at ambient temperature and δ-Bi_2_O_3_ at high temperatures are stable, whereas the remaining four are metastable phases [23]. The synthesis of Bi_2_O_3_ NPs can be done through various methods such as chemical vapour deposition (CVD) [24], precipitation method [25], electrodeposition [26], microwave-assisted synthesis [27], microemulsion method [28], sol-gel methods [29], hydrothermal synthesis [30] and so on. The aggregation behaviour of the metal oxide nanoparticles limits its application and the only solution to reduce this is to modify the surface of the nanoparticles. The surface modification of the synthesized Bi_2_O_3_ NPs should be performed for improving the dispersion, surface activity, and biocompatibility of Bi_2_O_3_ NPs, thereby enhancing their physicochemical properties [31]. Herein, we make use of functionalized β-cyclodextrin for the surface modification of Bi_2_O_3_ NPs.

β-cyclodextrin (β-CD), a cyclic oligosaccharide, possesses a molecular structure that has a hydrophobic core chamber and a hydrophilic outside [32]. Based on chemical processes, a variety of substances may be “grabbed” by the cyclodextrin cavity to create inclusion complexes [33]. There have been reports of nanoparticles’ agglomeration being prevented, and their colloidal and thermal stability being increased, by grafting a cyclodextrin polymer onto them [34,35]. The ability to do this has led to the widespread usage of CDs in a variety of fields, including catalysis, pharmaceutical science, separation technology, and many others [36,37,38,39]. For the modification of β-cyclodextrin, biotin, a water-soluble vitamin, can be used. Biotin has special functions in chromatin structure, epigenetic gene control, and cell signaling [40].

Accordingly, the present work is directed towards an efficient method for the surface modification of Bi_2_O_3_NP using biotinylated β-cyclodextrin. We described the experimental detail for the preparation of Bi_2_O_3_ NPs using PVA (poly vinyl alcohol) as the reductant, in order to obtain the NPs with small particle size and size distribution that is needed for biomedical application. The developed biocompatible system was characterized by UV-Vis, FT-IR, XRD, SEM, and TEM. The antibacterial and anticancerous activity of the surface-modified system was also evaluated. The zeta potential of beta-CD, bismuth nanoparticles, and bismuth nanoparticles with biotinylated beta-CD was determined by the dynamic light scattering (DLS) method.

## 2. Results and Discussion

### 2.1. Characterization of Bi_2_O_3_ NPs

FTIR spectrum of Bi_2_O_3_ gave the following data: 1380 cm^−1^ and 2352 cm^−1^: ν_C-O_ (vibr), 421, 540 and 844 cm^−1^: ν_Bi-O_ (vibr) (Figure 1a). The XRD pattern of Bi_2_O_3_ nanoparticles shows strong and sharp peaks, indicating that the synthesized nanoparticles are in the crystals phase as represented in Figure 1b. The peaks are in support of the monoclinic crystals phase of Bi_2_O_3_ based on COD No. 96-152-6459 at the 2θ value of 23.91°, 27.27°, 30.22°, 32.90°, 38.07°, 39.74°, 48.82°, 56.16°, and 64.63° which are associated with the planes (102), (120), (012), (121), (131), (122), (113), (222), and (104). The crystallite size of Bi_2_O_3_ NPs is calculated by the Debye Scherrer equation and obtained in the range of 12–16 nm. Figure 1c is the UV-Vis DRS spectrum of Bi_2_O_3_ NPs which also confirms the formation of Bi_2_O_3_ NPs by exhibiting absorption peaks at 283 nm and 268 nm.

The SEM gives information about the surface morphologies and particle sizes of the prepared Bi_2_O_3_ sample. The SEM images of Bi_2_O_3_ under different magnifications are presented in Figure 1d. The Bi_2_O_3_ NP has a spherical shape in morphology with an average diameter of 12–16 nm.

Figure 1e depicts the TEM image of Bi_2_O_3_ NPs, and it is clear that the synthesized monoclinic Bi_2_O_3_ NPs possess a spherical shape with a crystalline diameter in the range of 12–16 nm. The Selected Area Electron Diffraction (SAED) patterns of Bi_2_O_3_ NPs are shown in Figure 1f and the ring diffractogram agreed with the crystalline structure of monoclinic Bi_2_O_3_ NPs. Each ring’s diameter reflects the interplanar distance of a plane system that is present in the sample. The particle size obtained by TEM analysis was highly consistent with that obtained using XRD analysis.

### 2.2. Characterization of the Surface-Modified Bi_2_O_3_ Nanoparticles Using Biotinylated β-Cyclodextrin

The FT-IR spectrum of the BiONP-β-CD-Biotin system gave the following data (Figure 2a): 3295 cm^−1^: ν_O-H_ (str) of β-CD, 2922 cm^−1^: ν _C–H_ (aliph, str) of CH_2_, 1722 cm^−1^: ν _C=O_ (str) of COOR,1624 cm^−1^: ν _C=C_ (aliph, str), 1255 cm^−1^: ν _C–H_ (overtone, str) of biotin, 1481 cm^– 1^: ν _C–H_ (str) of biotin, 1020 cm^− 1^: ν _C–O_ (str) of biotin, 423 cm^−1^, 566 cm^−1^ and 851 cm^−1^: ν _Bi–O_ (vibr). The basic nature and size of the surface-modified Bi_2_O_3_ nanoparticles using biotinylated β-cyclodextrin were interpreted based on the XRD analysis and are shown in Figure 2b. The peaks are in support of the fact that the monoclinic crystals phase of Bi_2_O_3_is conserved in the final surface-modified system and the 2θ values obtained are in association with the planes (102), (120), (012), (121), (131), (122), (113), (222), and (104). The low-intensity peaks which are obtained below 30° are due to the β–CD in the final surface-modified system.

Using SEM, surface morphology studies of the functionalized β-CD system on the surface-modified Bi_2_O_3_ NPs were carried out in Figure 2c. The surface-modified Bi_2_O_3_ NPs with the functionalized β-CD system were found to have a well-dispersed morphology. Figure 2d displays transmission electron microscope (TEM) and Figure 2e the SAED images of surface-modified Bi_2_O_3_ NPs with the functionalized β-CD system, which shows that the particle in the surface-modified product retains its original shape, surface morphology, and particle size.

### 2.3. Antimicrobial Studies

The gram-positive bacterial strains (*S. pneumoniae* and *S. aureus*) and two gram-negative (*P. aeruginosa* and *K. pneumoniae*) strains were used to analyse the bacterial resistance by the agar diffusion test. Measurements were done five times for each strain. In all tests, the control did not achieve an inhibition zone. The diameter of the zone of inhibition for each bacterial strain was determined from the mean value of five replicates. Results are represented as the mean of five replicates ± standard deviation (Table 1). The average diameter of inhibition zones for surface-modified Bi_2_O_3_ NPs against *S. pneumoniae, S. aureus, P. aeruginosa,* and *K. pneumonia* are 14.26 mm Figure 3a, 17.54 mm Figure 3b, 15.31 mm, Figure 3c, and 12.23 mm Figure 3d, respectively.

### 2.4. In Vitro Cytotoxicity Studies

In vitro cytotoxicity of three different cancer cell lines; Hep G2, MCF-7, and A549 were evaluated using MTT assays and the results were illustrated as a graphical plot in Figure 4 and Table 2. For this purpose, the cancer cell lines were exposed to the BiONP-β-CD-Biotin system at various concentrations (6.25, 12.5, 25, 50, and 100 μg/mL) for 24 h. In all three cases, the cell viability declined with a rise in the concentration of th surface-modified Bi_2_O_3_ NPs with biotinylated β-cyclodextrin. The IC_50_ value of surface-modified Bi_2_O_3_ NPs with biotinylated β-cyclodextrin reported against Hep G2, MCF-7, and A549 cell lines are 76.743, 62.336, and 67.363 µg/mL respectively and it was determined using ED50 PLUS V1.0 Software. All studies were conducted in triplicate, with results expressed as Mean ± SE. A one-way analysis of variance (ANOVA) and Dunnets test were used to analyse the results. *p* < 0.001 in comparison to the control group was statistically significant.

### 2.5. Thermal Studies

The relationship between the temperature change and mass loss is expressed using DSC (Differential Scanning Calorimetry) and the DSC plot of the surface-modified Bi _2_O_3_ NP system is shown in Figure 5 [41,42,43]. On increasing the temperature from room temperature, in the DSC plot, an endothermic peak was observed where the onset temperature, the peak temperature, and the end set temperature are at 54.22 °C, 78.19 °C, and 104.23 °C, respectively. The corresponding area under the curve is 1393.97 mJ and the ΔH value, 139.39 J/g, is responsible for the formation of more thermally stable products.

### 2.6. Zeta Potential Analysis

The zeta potential of beta-CD, bismuth oxide nanoparticles, and bismuth oxide nanoparticles with biotinylated beta-CD was determined by the dynamic light scattering (DLS) method using a nanoparticle analyser (Horiba, nanopartica SZ 100). The zeta potential of beta-CD, Bi_2_O_3_ NPs, and Bi_2_O_3_ NPs with biotinylated beta-CD was obtained as -26.30 mV, 45.10 mV, and 41.20 mV respectively (Figure 6). The value of ±30 mV is normally used to assume the stability of NPs; Zeta potential values more than +30 mV and less than −30 mV represent stable conditions, but values between −30 mV and +30 mV show unstable conditions [44]. Zeta potential is thus indicative of the probable physical stability of a formulation [45]. The positive zeta potential value of bismuth nanoparticles with biotinylated beta-CD was due to the presence of biotin and bismuth nanoparticles. The reduction in the zeta potential value of bismuth nanoparticles with biotinylated beta-CD compared to that of bismuth was due to the effective grafting of negatively charged beta-cyclodextrin molecules on the surface of biotinylated bismuth nanoparticles.

## 3. Experimental

### 3.1. Materials and Methods

Bismuth nitrate alkaline Bi_5_O(OH)_9_(NO_3_)_4_ was used as a precursor. All of the chemicals and solvents used in the investigation were of analytical grade. Sodium-hypophosphite (NaH_2_PO_2_.H_2_O), sodium hydroxide (NaOH), nitric acid (HNO_3_), poly vinyl alcohol (PVA), ethanol, biotin, cyclodextrin, and DMSO are purchased from Nice chemicals. UV-Vis spectra were captured using a Shimadzu UV-visible NIR spectrophotometer (Shimadzu Corporation, Koyoto, Japan) operating between 190 and 1100 nm, and X-ray diffraction (XRD) measurements were performed using a Rigaku Miniflex-600 X-ray diffractometer with CuKα radiation in a θ–2θ configuration. The FT-IR spectra were captured using a Shimadzu-400 spectrophotometer with a scanning range of 4000–400 cm^−1^. A HITACHI S-4200 scanning electron microscope operating at 20 kV and a JEOL JEM-2100 transmission electron microscope (TEM) were utilized to examine the morphology and size of the Bi_2_O_3_ NPs and surface-modified products. The zeta potential of beta-CD, bismuth nanoparticles, and bismuth nanoparticles with biotinylated beta-CD was determined by the dynamic light scattering (DLS) method using a nanoparticle analyser (Horiba, nanopartica SZ 100, Horiba Ltd., Minami-ku, Koyoto, Japan).

### 3.2. Synthesis of Bi_2_O_3_ NPs

Initially, 0.052 g of PVA and 5 mL of 1 M Bi(NO_3_)_3_ were dissolved in concentrated nitric acid and 5 mL of water was added to the beaker at room temperature. Then, 40 mL of 5 M NaH_2_PO_2_.H_2_O was added. Dropwise additions of 0.52 g of NaOH in 70 mL are added to the solution until the pH reaches 2.5, followed by 20 min of stirring at room temperature. After transferring the beaker to an 80 °C temperature-controlled oil bath, stirring continued for an additional one hour and thirty minutes. After the reaction was complete, the black precipitate was filtered and repeatedly washed with distilled water and ethanol to eliminate contaminations and byproducts.
2Bi(NO_3_)_3_ + 3NaH_2_PO_2_ + 3H_2_O → Bi_2_O_3_ + 3NaH_2_PO_3_ + 6HNO_2_

### 3.3. Synthesis of Surface-Modified Bi_2_O_3_ Nanoparticles Using Biotinylated β-Cyclodextrin

By esterifying the -OH group of β-CD with the -COOH group of biotin by DCC coupling, β-cyclodextrin was made to be biotinylated (Steglich esterification). To obtain biotinylated β-CD surface modified Bi_2_O_3_ NP, about 1 gm of biotinylated β-CD and 0.25 gm of Bi_2_O_3_ NP were dispersed in 10 mL chloroform separately. Both solutions were mixed and stirred continuously for 3 h. After the stirring, the solution was allowed to dry and the obtained powder is the biotinylated β-CD modified bismuth oxide nanoparticle. Figure 1 schematically depicts the methods involved in creating surface-modified Bi_2_O_3_ nanoparticles using biotinylated β-cyclodextrin.

### 3.4. Antimicrobial Activity

The antibacterial activity of the surface-modified system is evaluated by applying the agar well diffusion method. The antimicrobial activity of the BiONP-β-CD-Biotin system was evaluated against two gram-positive (*Streptococcus pneumoniae* and *Staphylococcus aureus*) and two gram-negative (*Pseudomonas aeruginosa* and *Klebsiella pneumoniae*) bacterial strains. To inoculate the agar, a volume of the microbial inoculum is dispersed throughout its whole surface. Then, a volume of extract solution (20–100 μL) is injected into the well by creating a hole using a sterile cork borer. The test bacterial strain is then placed on it and the incubation procedure is repeated. The antimicrobial substance inhibits the growth of the tested microbial strain as it diffuses throughout the agar media. The diameters of the growth inhibition zones are then measured.

### 3.5. In Vitro Cytotoxicity Assays—MTT Assay

For cytotoxicity studies of chemicals and drug screening, in vitro cell viability and cytotoxicity studies with cultivated cells are often performed [46]. Currently, these tests are utilized in cancer research to evaluate the cytotoxicity and growth inhibition of possible cancer treatment candidates. In cell viability and cytotoxicity research, numerous cell activities, such as cell membrane permeability, ATP synthesis, cell adhesion, coenzyme creation, enzyme activity, and nucleotide absorption activity are measured [47]. MTT (3-(4,5-dimethylthiazol-2-yl)-2-5-diphenyltetrazolium bromide) assay is the most widely used colorimetric method for assessing cytotoxicity or cell viability [48]. By evaluating the activity of mitochondrial enzymes such as succinate dehydrogenase, this assay primarily analyses cell viability by measuring the efficiency with which cells carry out the mitochondrial function [49]. NADH transforms MTT into a purple formazan for this assay. Light absorbance at a particular wavelength can be used to calculate the amount of this product. The MTT method is widely used because it is simple to use, safe, and well-reproducible. In the present work, in vitro cytotoxicity of the BiONP-β-CD-Biotin system was assessed against cancer cell lines; Hep G2 (Human liver cancer), MCF-7 (Human breast cancer), and A549 (Human lung cancer). The study was conducted at the Center for Research on Molecular and Applied Sciences *Pvt Ltd.*, Biogenix, DNRA 41, Thiruvananthapuram, Kerala, India. The cancer cell lines were cultured in a 25 cm^2^ tissue culture flask containing DMEM supplemented with 10% FBS, L-glutamine, sodium bicarbonate, and an antibiotic solution containing Penicillin (100 U/mL), Streptomycin (100 μg/mL), and Amphoteracin B (2.5 μg/mL). In a humidified, 5% CO_2_ incubator, cell lines were maintained at 37 °C. A 96-well tissue culture plate, 5 × 10^3^ cells/well were sown and cultured at 37 °C in a humidified 5% CO_2_ incubator. After 24 h, the growth medium was removed, surface modified Bi_2_O_3_ NPs with biotinylated β-cyclodextrin in DMEM were five times serially diluted by two-fold dilution (100 µg, 50 µg, 25 µg, 12.5 µg, 6.25 µg in 500 µL of DMEM), and each concentration of 100 µL was added in triplicates to the respective wells and incubated at 37 °C in a humidified 5% CO_2_ incubator. Non-treated control cells were also maintained. After 24 h of incubation, the sample content was withdrawn from the wells and 30 µL of reconstituted MTT solution was added to all test and cell control wells. The plate was then gently shaken and incubated at 37 °C in a humidified 5% CO_2_ incubator for four hours. After the incubation period, the supernatant was removed, 100 µL of MTT solubilization solution was added, and the wells were pipetted up and down gently to dissolve the formazan crystals. At a wavelength of 540 nm, absorbance values were measured using a microplate reader. Using the formula, the percentage of growth inhibition was computed:$$\%\;\mathrm{of}\;\mathrm{viability}=\frac{\mathrm{Mean}\;\mathrm{OD}\;\mathrm{samples}}{\mathrm{Mean}\;\mathrm{OD}\;\mathrm{of}\;\mathrm{control}\;\mathrm{groups}}\times 100$$

All studies were conducted in triplicate, with results expressed as Mean ± SE. A one-way analysis of variance (ANOVA) and Dunnets test were used to analyse the results. *p* < 0.001 in comparison to the control group was statistically significant.

## 4. Conclusions

Recently, the surface modification of metal oxide nanoparticles arises as a new class in nanotechnology with wide applications. Upon surface modification, the nanoparticles become stabilized against agglomeration and this type of modification also renders them compatible with another phase. In this article, we prepared Bi_2_O_3_ NPs with modified surfaces using the functionalized β-CD system, a system that is completely biocompatible. Using XRD, FT-IR, UV-Vis, TEM, and SEM respectively, all the acquired products were characterised. The thermal events that occurred in the final product were carried out using DSC. In this work, we also investigated the antibacterial and anticancerous influence of the BiONP-β-CD-Biotin system using the bacterial strains *S. pneumoniae*, *S. aureus*, *P. aeruginosa,* and *K. pneumonia,* and the cancerous cells Hep G2, MCF-7, and A549. The zeta potential of beta-CD, bismuth nanoparticles, and bismuth nanoparticles with biotinylated beta-CD was determined by the dynamic light scattering (DLS) method, which is indicative of the probable physical stability of the BiONP-β-CD-Biotin system.

## Data Availability

All relevant data are included in the article.

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
