# Peer review of "Surface Modification of Bi2O3 Nanoparticles with Biotinylated β-Cyclodextrin as a Biocompatible Therapeutic Agent for Anticancer and Antimicrobial Applications"

_molecules, 2023, doi:10.3390/molecules28083604_

Round 1
Reviewer 1 Report
From my perspective, this is an interesting work; however, the following issues have to be addressed carefully.
- In the abstract section, you need to focus more on quantitative information, not qualitative ones.
- The authors did not explain the novelty and significance of their work in the introduction part. Indeed, the introduction part is not cohesive. Topics change from sentence to sentence. The authors should follow the funnel procedure. The funnel technique for writing the introduction begins with generalities and gradually narrows your focus until you present your thesis.
- The first sentence of the introduction "Inorganic nanoparticles (NPs) have achieved great significance in recent times for their potential to be used in biological and therapeutic applications." also needs the following references: https://doi.org/10.1016/j.ceramint.2021.10.088; https://doi.org/10.1016/j.surfcoat.2019.01.002
- Throughout the 'materials and methods section the authors must always state the number of samples tested and/or the number of analyses made per sample. The number of repetitions is important to validate the results' scientific accuracy. A result that was obtained once with no scientific significance.
- In Results and Discussion, in each section, authors have reported and discussed many aspects, thus the key message from the section is not clearly understood. Authors are encouraged to reduce the texts by reporting the key findings, then having a discussion of the results with the relevant literature.
- Please unify the bibliographical references. Some references are separated by commas, others by hyphens. Please choose which one to use.
- There are some formatting and spelling errors in this manuscript, a full-text check should be performed.
Author Response
Reviewer #1:
Q1:In the abstract section, you need to focus more on quantitative information, not qualitative ones.
R1: Based on the reviewer suggestion, in the abstract section more quantitative informations are added.
Q2: The authors did not explain the novelty and significance of their work in the introduction part. Indeed, the introduction part is not cohesive. Topics change from sentence to sentence. The authors should follow the funnel procedure. The funnel technique for writing the introduction begins with generalities and gradually narrows your focus until you present your thesis.
R2: As suggested significance of the work is described in the introduction part.
Q3:The first sentence of the introduction "Inorganic nanoparticles (NPs) have achieved great significance in recent times for their potential to be used in biological and therapeutic applications." also needs the following references: https://doi.org/10.1016/j.ceramint.2021.10.088; https://doi.org/10.1016/j.surfcoat.2019.01.002
R3: The references mentioned by the reviewer are added in the first sentence.
Q4: Throughout the 'materials and methods section the authors must always state the number of samples tested and/or the number of analyses made per sample. The number of repetitions is important to validate the results' scientific accuracy. A result that was obtained once with no scientific significance.
R4: In antibacterial studies, measurements were done by keeping five replications. The results of the inhibition zone observed was represnetd as mean±standard deviation. In vitro anticancer studies were conducted in triplicate, with results expressed as mean±standard deviation.
Q5: In Results and Discussion, in each section, authors have reported and discussed many aspects, thus the key message from the section is not clearly understood. Authors are encouraged to reduce the texts by reporting the key findings, then having a discussion of the results with the relevant literature.
R5: As suggested by the reviewer, the result and discussion section was reframed and added relevant literatures.
Q6: Please unify the bibliographical references. Some references are separated by commas, others by hyphens. Please choose which one to use.
R6: Corrections are carried out in the bibliographical references.
Q7: There are some formatting and spelling errors in this manuscript, a full-text check should be performed.
R7: A full check on the manuscript has been carried out.
Reviewer 2 Report
The manuscript ‘Surface Modification of Bi2O3 Nanoparticles with Biotinylated β-Cyclodextrin as a Biocompatible Therapeutic Agent for Anticancer and Antimicrobial Applications’ is a very interesting paper to describes a new route to the surface modification of bismuth oxide nanoparticles (Bi2O3 NPs) using functionalized beta-Cyclodextrin (β-CD) as a biocompatible system. The manuscript provides a large amount of data. Congratulations to the authors.
The manuscript is complete and the different sections of the article are well balanced and adequately supported by the data provided. The manuscript is clear and answers the objectives. The manuscript is appropriate from the aims and scope of the journal and is potentially publishable “Molecule” with some minor changes.
1. The introduction section need to be improved. The authors can strengthen Justification of work by reading and adding some recent review and research article.
2. Add hypothesis in to introduction section
3. The quality of Figure 1 to 3 is very poor. need to improve
4. Statistical analysis need to improve.
5. The whole article needs to be checked critically for typos and grammar errors
6. All the references should be according to Journal guidelines.
Author Response
Reviewer #2:
Q1: The introduction section need to be improved. The authors can strengthen Justification of work by reading and adding some recent review and research article.
R1: Based on the suggestion of the reviewer, introduction section has been improved.
Q2: Add hypothesis in to introduction section
R2: Hypothesis are added in the introduction.
Q3:The quality of Figure 1 to 3 is very poor. need to improve.
R3: Fig 1-3 are redrawn based on the opinion of the reviewer.
Q4: Statistical analysis need to improve.
R4: The in vitro cytotoxicity studies, one-way analysis of variance (ANOVA) and Dunnets test were conducted. The results are incorporated in paper.
Q5:The whole article needs to be checked critically for typos and grammar errors.
R5:The article has been checked for typos and grammar errors.
Q6: All the references should be according to Journal guidelines.
R6:.References is corrected based on journal guidelines.
Reviewer 3 Report
The abstract and introduction section are too long, with lots of information are meaningless. The experimental steps contain too many explanations of the experimental principles, making this study more like a dissertation. The entire experimental design is overly simplistic and crude. The particle size and surface potential of the prepared nanoparticles were not characterized in the whole paper, and the grafting rate of biotin was lacking. In this paper, nanoparticles without biotin modification were not prepared as control materials to better illustrate the biological functions brought by biotin modification. It is recommended to refer to the following literature and carry out more antimicrobial performance evaluation. 10.1016/j.carbpol.2022.119130; ACS Appl. Mater. Interfaces 2020, 12, 22479;
Author Response
Reviewer #3:
Q1:The abstract and introduction section are too long, with lots of information are meaningless.
R1:As suggested by the reviewer, introduction section has been modified.
Q2:The experimental steps contain too many explanations of the experimental principles, making this study more like a dissertation. The entire experimental design is overly simplistic and crude.
R2: As suggested by the reviewer, we modified the experimental principles and incorporated the necessary corrections.
Q3:The particle size and surface potential of the prepared nanoparticles were not characterized in the whole paper, and the grafting rate of biotin was lacking.
R3: As suggested by the reviewer, the particle size of nanoparticle is calculated from XRD data using Debye-Scherrer’s relation and obtained in the range of 12-16 nm.The surface potential of prepared nanoparticles were not measured, due to the unavailability of the instrument.
Q4:In this paper, nanoparticles without biotin modification were not prepared as control materials to better illustrate the biological functions brought by biotin modification.
R4: The anticancer and antibacterial studies for Bi2O3 nanoparticles without biotin modification were already did and the results showed very less values. So the results were insignificant to add in the manuscript. And the surface modification of nanoparticles improved the anticancer and antimicrobial activity.
Q5:. It is recommended to refer to the following literature and carry out more antimicrobial performance evaluation. 10.1016/j.carbpol.2022.119130; ACS Appl. Mater. Interfaces 2020, 12, 22479;
R5: As suggested by the reviewer, we referred the following literature and added it in reference.
Round 2
Reviewer 1 Report
The authors justified the queries made by this reviewer. I think the revised version may be accepted in its current form.
Author Response
Thank you for your positive comments
Reviewer 3 Report
The authors have answered my question well, and the paper can be considered for publication now.
Author Response
Thank you for the review activity and comments